# Protein Profiling of *Psittacanthus calyculatus* during Mesquite Infection

**DOI:** 10.3390/plants12030464

**Published:** 2023-01-19

**Authors:** Montserrat Aguilar-Venegas, Elizabeth Quintana-Rodríguez, Víctor Aguilar-Hernández, Claudia Marina López-García, Efraín Conejo-Dávila, Ligia Brito-Argáez, Víctor M. Loyola-Vargas, Julio Vega-Arreguín, Domancar Orona-Tamayo

**Affiliations:** 1Ciencias Agrogenómicas, Escuela Nacional de Estudios Superiores, Unidad León, UNAM, León CP 37684, Guanajuato, Mexico; 2Medio Ambiente y Biotecnología, CIATEC, A.C., León CP 37545, Guanajuato, Mexico; 3Unidad de Bioquímica y Biología Molecular de Plantas, CICY, A.C., Mérida CP 97205, Yucatán, Mexico; 4Unidad Profesional Interdisciplinaria de Ingeniería Campus Guanajuato, Instituto Politécnico Nacional, Silao de la Victoria CP 36275, Guanajuato, Mexico

**Keywords:** *Prosopis laevigata*, infective stages, germination, seedling establishment, haustorium development

## Abstract

*Psittacanthus calyculatus* is a hemiparasite mistletoe that represents an ecological problem due to the impacts caused to various tree species of ecological and commercial interest. Although the life cycle for the *Psittacanthus* genus is well established in the literature, the development stages and molecular mechanism implicated in *P. calyculatus* host infection are poorly understood. In this study, we used a manageable infestation of *P. laevigata* with *P. calyculatus* to clearly trace the infection, which allowed us to describe five phenological infective stages of mistletoe on host tree branches: mature seed (T1), holdfast formation (T2), haustorium activation (T3), haustorium penetration (T4), and haustorium connection (T5) with the host tree. Proteomic analyses revealed proteins with a different accumulation and cellular processes in infective stages. Activities of the cell wall-degrading enzymes cellulase and β-1,4-glucosidase were primarily active in haustorium development (T3), while xylanase, endo-glucanase, and peptidase were highly active in the haustorium penetration (T4) and xylem connection (T5). Patterns of auxins and cytokinin showed spatial concentrations in infective stages and moreover were involved in haustorium development. These results are the first evidence of proteins, cell wall-degrading enzymes, and phytohormones that are involved in early infection for the *Psittacanthus* genus, and thus represent a general infection mechanism for other mistletoe species. These results could help to understand the molecular dialogue in the establishment of *P. calyculatus* parasitism.

## 1. Introduction

Parasitic plants are angiosperms that need to attach to other plants that are susceptible to being parasitized, and obtain nutrients from them through a specialized structure called a haustorium [1]. More than 4750 of the 374,000 reported flowering plants have a parasitic habit [2]. The parasitic plant diversity depends on the habit and site of attachment on the host and their ability to acquire nutrients derived from the host metabolism [3,4]. In this sense, some parasitic plants have the capacity to obtain some of their nutrients from photosynthesis (hemiparasites), and additionally, they supplement their nutrition from the nutritional components derived from the host plant. Classic examples of hemiparasitic plants are the mistletoes, that are aerial parasitic plants that are composed of more than 1500–1600 species worldwide [5,6,7,8], and are found in diverse forms on host plants in tropical zones from South and Central America, Africa, New Zealand, and Australia [5,9,10,11].

Specifically, the Loranthaceae family mistletoes, comprising 76 genera and around 1000 species, represent one of the largest families in the Santalales order [5]. The Santalales order comprises members that need to be latched onto a host plant through a modified root system called a haustorium to acquire water, mineral nutrients, and organic compounds for development [7]. They are widely distributed in many parts of the world, and have turned into a serious forest pest by causing shoot die-off and a drastic reduction in tree vitality, biomass yield, and flowering and fruiting, and increase susceptibility to insects and phytopathogens in the host [12,13,14,15,16].

The *Psittacanthus* genus is the most striking hemiparasite group, that comprises around 120 species, that are distributed from North America to South America [17,18]. Different mistletoe species are found in 25 Mexican states situated in the north, central, and southern regions [8,19], and one of these mistletoes is *Psittacanthus calyculatus* (DC.) G. Don (Loranthaceae), a Mexican mistletoe, with a high prevalence in the central region of Mexico and that infests more than 65% of mesquite (*Prosopis laevigata*), an endemic tree in this region [8], and additionally, it can infect and endanger tree species such as those from the genera *Citrus* and *Persea* or the families Fabaceae and Coniferae [19,20,21]. *P. calyculatus* produces showy bright yellow- and orange-colored flowers, with the capacity to produce high concentrations of floral nectar secretion, volatile organic compound emission, and high amounts of petal carotenoids; those traits serve to engage and manipulate pollinator visitors, thus to ensure reproduction and subsequent propagation [8,22].

The seeds of the mistletoe are dispersed by frugivorous birds that perch in trees and deposit the seeds by regurgitation or defecation [23,24]. Once the seed is attached to the host branch, with the suitable environmental conditions present in the host tree niche, such as temperature, light, and humidity, *P. calyculatus*’s life cycle starts with the germination of mature seeds, initiating the penetration of host branches and haustorium formation that allows it to take nutrients from the host [1,22,25]. The haustorium is a specialized invasive organ that, after attachment to the host tissue, penetrates and establishes a connection with the vascular bundle [26,27], interfering strongly with host physiological processes, i.e., development, growth, and reproduction, that gradually drives defoliation and death [28,29]. However, in the current state of the art related to parasitic plant infection, there are only studies that describe the chemical and molecular interaction in parasitic plants other than mistletoes, therefore the information must be adapted. In that sense, some results have described the initiation of the haustorium development, which includes chemical and physical interactions between haustorium and host tissues. For example, the penetration process is well documented in root parasitic species such as *Cuscuta reflexa* and *C. jalapensis*, *Orobranche, Striga*, and *Phelipanche* [30,31,32,33,34]. In addition, *Orobranche aegyptiaca* releases various cell wall-degrading enzymes such as cellulases, hemicellulases, and polygalacturonases that break down lignocellulolytic material associated with cellulose, and proteases that degrade proteins of the host cortex for its successful penetration into the host root [35,36]. Phytohormone accumulation is important for haustorium development. Diverse parasitic hormones have been anticipated in the site of infection from gene expression analysis related to accumulation of auxins in the early stages of the haustorium development [36,37,38,39,40]. Nevertheless, successful infection of *P. calyculatus* in a host tree is dependent on the establishment mechanisms in phenological stages such as germination and growth of seedlings, as well as metabolic signatures in haustorium development, enzymes involved in bark layer degradation and xylem connection, and molecular and genetic factors that inhibit the host immune system [34,41,42,43,44,45,46]. Therefore, molecular analyses on the development of the haustorium are necessary, involving scanning biomolecules such as phytohormones, proteins, and enzymes involved in haustorium development and thus in the establishment on the host.

We characterized the differential infection stages between *P. calyculatus* and its mesquite host tree (*Prosopis laevigata*), and explored the infective stage proteomes of: mature seed, germination, seedling establishment, haustorium penetration, and connection with the host xylem tissue. Furthermore, we quantified metabolic biomarkers such as chlorophyll contents and protease activities during seed-to-seedling transition. Additionally, quantification of the auxin and cytokinin level and cell wall-degrading enzymes such as cellulases, xylanases, endo-1,4-β-glucanases, and β-1,4-glucosidases was performed to understand the molecular mechanism in the invasion into the host.

## 2. Results

### 2.1. The Study System

*P. calyculatus* is a hemiparasitic mistletoe that infects diverse trees, including *P. laevigata* (mesquite). Due to its high multi-infective capacity, its germination and infection processes are complex and it is necessary to understand the processes that orchestrate the infection. In that aspect, Geils and Hawksworth [47] describe the phenology, development, and the life cycle of the *Psittacanthus* genus. To establish the first infective stages based on Geil’s model, we performed and adapted in an artificial *P. calyculatus* seed inoculation on the host branch to monitor the different stages of germination and infection, that allowed us to analyze the molecular and biochemical changes that took place during the first six months of the infection. In that sense, five stages were selected according to anatomical features of exophytic parts of the mistletoe body and establishment, and the chronology of the development of each of them was documented as well.

### 2.2. Germination and Infection Development

After the seeds were placed manually (T1; mature seeds without epicarp) onto the branch (Figure 1A), the embryonic cotyledons (T2) were released (≈7–14 days; dai) from the exocarp (seed cover; Figure 1B), then the cotyledon started to open “like a star” (Figure 1C). In this stage, the prehaustorium development took place (T3) at 21–35 days (dai) (Figure 1M). The next growth stage, organogenesis (T4), was characterized by cotyledons opening fully and the first young leaves (Figure 1D,I). At that time, from 60–90 days after the fruit/seed germination process, the haustorium had already penetrated the bark and xylem, reaching the phloem of the host branch (Figure 1N). Complete seedlings with true leaves were found from 4–6 months after seeds were inoculated (Figure 1E,J). Given that those morphological changes occurred on the host, the cortex of the host was inspected during material collection. We noted that bark damage began at the immediate point of contact of the prehaustoria (T3) with the host branch (Figure 1H,M). Subsequently, at stage T4, the host bark on the branch continued to sustain damage since bark breakdown and intrusive penetration by the pressure and connection (T5) of the haustorium with the xylem of the host branch were noted (Figure 1N). Next, the samples of each stage were dissected and the endophytic system was analyzed by a cross section. In comparison with mature seed, the embryo size was increased at T2 stage (Figure 1L). Then, when the cotyledons open, holdfast and haustorium meristem were formed (Figure 1M). During true leaf development, the intrusive organ reached the host cambium (Figure 1N) and the haustorium invaded the sapwood, and finally, the tissue was connected with the host xylem (Figure 1O).

In summary, we found that germination and seedling establishment of mistletoe are synchronized events that involve early invasion stages through haustorium development and the haustorium’s connection with vascular tissue of the host branch, culminating with the host tissue invasion. Five invasive stages were identified: (T1) mature seed, (T2) formation of holdfast, (T3) haustorium activation, (T4) intrusive organ development/host penetration, and (T5) haustorium connection with xylem tissue.

### 2.3. Proteomic Dynamics during Invasive Stages

We followed the development of the five invasive stages and the leaf protein content as a reference pattern by extracting total proteins and separating them via by one- and two-dimensional gel electrophoresis. The unidimensional SDS-PAGE exhibited approximately 20 protein bands per infective stage that ranged from 10–120 kDa (Figure 2).

Some predominant bands were shown in most of the stages, apparently having only qualitative differences. Next, to elucidate the proteins that changed pattern during the invasive process, a 2-DE in the pH 4–7 range was performed. Around 105 individual spots were selected and excised from the different gels for identification by LC-MS/MS. Peptide masses and MS/MS profiles were searched using the MS BLAST search engine to identify entries (Appendix A). However, only thirty-two proteins positively matched with entries in the NCBI database, which are shown in Table 1. The majority of those proteins represented different putative functionalities (Figure 3), and are enzymes involved in gene regulation (31.25% of the spots; e.g., RNA-binding proteins, DNA replication factor, and exonuclease family proteins), followed by cellular response (18.75%; e.g., RING finger domain, organ morphogenesis, absorption proteins). Further quantitatively important functional groups included proteins that are related to other metabolic processes such as metabolism (25.0%; e.g., DeSI-like protein is similar to a peptidase), development processes (9.37%; e.g., cell wall biosynthesis), cell division (9.37%; e.g., cyclin-D5-3-like and leucine-rich proteins), and other minor proteins with roles in phytohormone synthesis (3.12%; e.g., oxydoreductases) and photosynthesis (3.12%; e.g., glutamine aminotransferase, Figure 3).

The optical densities of the 33 protein spots were quantified using ImageJ software in Coomassie-stained gels to generate a heat map during each invasive stage of infection on the host branch (Figure 4A). Diverse protein spots were found to present accumulation in different mistletoe infection stages. For example, some proteins are grouped in section III of the heat map, and are found in almost all stages (Figure 4B) such as pectinesterase (11), and exonuclease family protein (31), leucine-rich repeat protein (32), and chaperone protein (20), which collectively are involved mainly in catabolism and gene regulation. On the other hand, proteins grouped in section II participate in diverse processes such as metabolism, hormonal pathway signaling, and gene regulation, and all of them were accumulated in T4 and T5 infective stages. Section I includes diverse proteins that decreased in the seed-to-seedling transition, in fact, DeSI-like protein (8), heat shock transcription factor A6B (9), U2 small nuclear ribonucleoprotein (5), SKP1-like protein (3), and PYK10-binding protein (4) were not present in stages T4 or T5 (Figure 4B). Nevertheless, other proteins such as zein-binding protein (10), sugar transporter (12), and RING/U-box superfamily protein (19) were only found in T1 and T5 infection stages. It is important to note that the proteins of the mistletoe leaves did not show accumulation comparable to the different proteins derived from the mistletoe infectious stages, but we found an accumulation of pectinesterase (11) in the mistletoe leaves.

### 2.4. Cell Wall-Degrading Enzymes during Host Invasion

The plant cell wall is a complex structure mainly composed of cellulose and xyloglucans polymers [48]. The host intrusion and penetration by the parasitic haustorium are the most important events, as the haustorium tissue needs to break down a complex network of lignocellulose. Cell wall weakening or loosening of host cortex tissue is a prerequisite for intrusive organ penetration after mistletoe fixation to the host [49,50,51]. Specific substrates were used and revealed four types of glycosyl hydrolases: cellulase-, endo-1,4-β-glucanase-, β-1,4-glucosidase-, and xylanase-like activities, from *P. calyculatus* in different invasive stages (Figure 5). Cellulase-like activity was detected in all infective stages (Figure 5A), and the activity increased significantly (*p* < 0.05) over stage T3 with a value of 5.04 μmol Glcmg^−1^min^−1^, and a reduction at T4 and T5 (3.6 μmol Glc mg^−1^min^−1^). The increase in endo-1,4-β-glucanase-like activity was shown to be statistically significant (*p* < 0.05) in stages T1 and T2 (0.4 Glc mg^−1^min^−1^), dramatically decreased in stage T3 (0.3 Glc mg^−1^min^−1^; *p* < 0.05), and was the highest in stage T4 (0.5 Glc mg^−1^min^−1^), decaying again in stage T5. A high activity of this enzyme was detected in the mistletoe leaves (0.6 Glc mg^−1^min^−1^; *p* < 0.05), compared with the other stages. β-1,4-glucosidase-like activity gradually increased (Figure 5C), reaching the highest at stage T3 (5.3 μmol Glc mg^−1^min^−1^; *p* < 0.05), decreased at stage T4 (2.6 μmol Glc mg^−1^min^−1^), and increased again at stage T5 (4.01 μmol Glc mg^−1^min^−1^), and the activity in the leaves was higher (7.3 μmol Glc mg^−1^min^−1^). By contrast, xylanase-like activity showed the highest activity at stage T1 with 2.5 μmol Glc/mg x min; *p* < 0.05 (Figure 5D), decreased at stages T2 and T3 (1.71 μmol Glc mg^−1^min^−1^; *p* < 0.05), and increased in stages T4 and T5 (2.3 μmol Glc mg^−1^min^−1^), and the activity in leaves was higher (2.7 μmol Glc mg^−1^min^−1^; *p* < 0.05).

### 2.5. Protease Activity and Chlorophyll Content

The proteomic profiles and subsequent MS/MS revealed a DeSI-like protein, a putative protease enzyme [52,53,54]. DeSI-like protein showed an increasing accumulation over the different infective stages (T1–T3; Figure 4), suggesting that the protein catabolism occurs during the invasive process. We then determined protease activity in the different invasive stages. Protease activity was maintained from stage T1 to T3 with an activity of 0.4 U/g protein, and significantly (*p* < 0.05) increased in stages T4 and T5 (0.58 and 0.54 U/g protein, respectively). Similar activities in mistletoe leaves were found (Figure 6). Next, the chlorophyll levels were analyzed in the infective stages as an indirect indicator of photosynthetic efficiency. As we expected, chlorophyll level in T1 (0.21 mg/g FW) decreased in T2–T4 (0.20–0.18 mg/g FW), and dropped dramatically in stage T4 (0.08 mg/g FW), and, surprisingly, the concentration presented similar levels between stage T1 and that in the mistletoe leaves (Figure 7).

### 2.6. Phytohormone Level during Invasive Stages

The success of host infection depends on the penetrative/invasive organ, in other words, haustorium development, which has been well studied in root parasitic plants. Auxin and cytokinin ratios play a pivotal role in haustorium formation [55]. Unfortunately, the mechanisms that regulate formation of the haustorium in *Psittacanthus* are still unknown. Here, during the invasive stages that comprise the haustorium development (Figure 1), we identified enzyme accumulations involved in auxin biosynthesis (tryptophan aminotransferase-related 2; 21) (Table 1). A phytohormonal profile of auxins and cytokinins that govern vegetative growth and haustorium development in *P. calyculatus* was determined by LC-HPLC analysis, and we found 4 auxins and 10 cytokinins in the five invasive stages. Auxins such as indole-3-acetic-acid (IAA), indole-3-acetyl-L-aspartic acid (IAA-L-Asp), indole-3-acetyl-L-glutamic acid (IAA-L-Glu), and indole-3-acetyl-L-alanine (IAA-L-Ala) and cytokinins such as dehydrozeatin (DZ), N6-furfuryladenine (kinetin), N6-isopentenyladenine (iP), N6-benzyladenine (BA), cis-zeatin-riboside (cZR), and trans-zeatin (tZ) were determined. Table 2 shows the fold change in the different invasive stages in comparison with the leaf for each auxin and cytokinin determined. IAA was detected in T1, T2, T4, and T5 stages, finding the highest level in the seed. IAA-L-Asp and IAA-L-Glu were found in all invasive mistletoe stages. Stage T1 had the highest content, and then a reduction was observed in seed-to-seedling transition, in fact, the lowest level was found in the leaf. IAA-L-Ala was not detected in either invasive stages or leaves. Cytokinins such as DZ were detected in T1 and increased in stage T3, and disappeared in T4 and T5 stages and in leaf. iP content was detected from T1 to T3 stages, showing a high concentration in stage T2. cZR was detected in a small amount in almost every sample analyzed, except in stage T4. tZ was only detected in stage T2 while kinetin and BA were not detected in either invasive stages or leaves (Table 2).

## 3. Discussion

Mistletoe infection on trees represents an important trait of interaction with a high ecological, evolutionary, and commercial importance. However, the biochemical mechanism by which *P. calyculatus* begins its infectious process in a mesquite tree remains to be elucidated. Our understanding of the protein and enzymatic mechanisms that control these adaptive responses remains rudimentary and needs to be deepened. The successful *P. calyculatus* infection in a host tree, depends of the different mechanisms of establishment such as seed germination, metabolic signatures in haustorium development, cell wall-degrading enzymes and xylem connection, as well as molecular and genetic factors that inhibit the host immune system. Information about mechanisms to establish the parasitism between mistletoe and a host tree is scarce. This study provides biochemical knowledge of the host infection of hemiparasite *P. calyculatus*, during six months of observation on a *P. laevigata* host. We reported the infective cycle of one of the most widely distributed mistletoes in the Americas, *P. calyculatus*, as well as a proteomic analysis, important biomarkers, and phytohormones as regulators of development processes during the invasion into the host.

### 3.1. Morphological Establishment of Mistletoe Invasive Stages

Our observations of infective stages of *P. calyculatus* were similar to those previously reported for the *Psittacanthus* genus [56]. The cycle starts when frugivorous birds disperse the mistletoe’s mature seeds onto other host trees. In the first weeks of progression of the infection, seed adhesion, germination, seedling establishment, and organogenesis of the first true leaves occur; during the next 4–7 months the vegetative growth continues (Figure 1). Thoday et al. [57] described the haustorium anatomy and penetration of many Santalales species, however, they only described the phenomena for seven of the twelve currently recognized parasitic plant lineages [58,59]. Hence, there are no data about the chronology of development and penetration of haustoria in *P. calyculatus*. To understand these processes, we focused firstly on morphological changes during seed adhesion, germination, seedling establishment, and true leaf organogenesis, which are development processes prior to haustorium connection [43,60,61]. We found and described the *P. calyculatus* tissue invasion process, characterized in five stages: 1) mature seed (T1), 2) formation of holdfast (T2), 3) haustorium activation (T3), 4) intrusive organ development (T4), and 5) haustorium connection to xylem tissue (T5) (Figure 1A–F), which correspond to different invasive infection stages of fructification, germination, seedling establishment, organogenesis of the leaves, and vegetative growth, respectively.

Mature *P. calyculatus* seed (Figure 1A,K) shows the typical berry structure: pulp or exocarp covered with a cuticle layer, mucilaginous mesocarp, endocarp, and chlorophyllous embryo with visible polycotyledonous structure that includes a hypocotyl [62,63]. Frugivorous birds deposit seed free of exocarp on host branches, where they remain glued onto the bark by viscin secretion from the mesocarp (Figure 1B,G). In the seed free of exocarp, cell division is activated at the tip of the hypocotyl, producing a dome-shaped holdfast [57], which was displayed by a transversal section from stage T2 (Figure 1L). Stage T3 includes cotyledon aperture and from the lower side of the holdfast, a primary haustorium emerges and grows, which shows a predominant sideways growth (Figure 1M) that could be a response to contact with the host bark [64]. In stage T4, the parasitic endophyte becomes passively embedded within some host dermal tissues [65,66]: i) external bark, a protection layer insulator, prevents losing moisture and mechanical damage, ii) phloem, tissue that transport organic nutrients to the rest of the tree, iii) cambium, tissue that provides partially undifferentiated cells for generation of new bark, wood, and xylem tissue. Is important to note that when the primary haustorium contacts the cambium tissue, it becomes a mature haustorium [67,68]. The outer bark is an inactive tissue, but phloem and cambium are not, and therefore mistletoe can take up nutrients from the phloem, but crossing these layers could activate the host immune response and lead to the arrest of the invasion [66,69,70]. Finally, in stage T5, the haustorium reaches xylem tissue to predate water, minerals to increase its growth and development (Figure 1O).

### 3.2. Protein Functions Involved in Seed Germination and Haustorium Development

Phenological data indicate that in *P. calyculatus* germination, seedling and haustorium development are necessary processes for the invasion into mesquite. This is because any change in the proteomic profile of seedling development could affect achievement of infection. The process of seedling establishment begins with seed germination and it is completed when the seedling becomes autotrophic. Germination was defined in orthodox seeds as those events that commence with the uptake of water by the quiescent dry seed and terminate with the elongation of the embryonic axis [41,71]. However, mistletoes that have recalcitrant seeds are able to germinate immediately after shedding, without a quiescent phase, and the exocarp needs to be removed by frugivorous birds to facilitate the germination process [72,73,74]. Deeks [75] defined the dwarf mistletoe germination as initiation of meristematic activity of the radicular apex until it reaches an obstruction, at which time holdfast tissue develops. In this context, we consider stages from T2 to T3 as germination stages as there is growth at the tip of seed and true leaves appear in stage T4 as an indicator that seedling establishment is complete. In that sense, during germination and seedling establishment there is rapid growth of the embryo, which implicates a high rate of cell division, and the cell cycle could be more highly regulated during the early infection process [76]. DNA replication licensing factor MCM2 [77] and WEB family proteins [78,79] are synchronized and prepare the cell cycle progression, respectively, while leucine-rich repeat-containing protein DDB is involved in chromatin and DNA dynamics [79]. On the other hand, DeSI-like (peptidase-like) protein is a protein involved in nitrogen catabolism [52,80,81], and these nitrogen derivates could be translocated and transformed into nitrogen reserves in seeds or be responsible for the emergence of seedlings [82].

Different phloem proteins were determined by a proteomic approach in a leguminous shrub, *Acacia cornigera*, and those proteins were quantified as having a high concentration and were translocated into specialized nectary tissues that secreted phloem assimilates [83], which shows that phloem proteins can be a specific protease targets and, together with structural proteins and lipoproteins, associated with different regions of the membrane and cell wall of the host tree. They might be degraded by the enzymatic proteolysis performed by the mistletoe haustorium, and the peptides and amino acids derived from proteolytic degradation will be translocated to the tissues of the parasite. In that aspect, we found that peptidase activity was active in mistletoe seeds and that activity increased when the mistletoe haustorium was connected with the host.

Proteins in seeds play a central role in dormancy release, metabolism resumption, and diverse processes [84]. In parasitic seeds, ABA drives the accumulation of late embryogenesis abundant (LEA) proteins, heat shock thermostable proteins (sHSPs), storage proteins (SSPs), endopeptidases, and carboxypeptidases, whose main functions are supported in early seedling growth and abiotic stress tolerance [85,86,87,88]. However, *P. calyculatus* seeds show recalcitrant traits such as high water content which cannot be conserved because they will not tolerate dehydration [88,89]. In that aspect, probably, this is the reason why we only detected four proteins related to ABA signaling or seedling development in *P. calyculatus* seeds: heat shock transcription factor A6B, SKP1 family protein 1, PYK10-binding protein 1, and zein-binding domain-containing protein. Heat shock transcription factor A6B is a class A HSF, extensively increased with salinity, osmotic, and cold stresses, that regulates the transcription in a DNA-dependent manner [90,91]. SKP1 protein may be a component of the E3 ubiquitin ligase SCF complex involved in ubiquitination. In Arabidopsis, the *SKP1* gene is expressed in the fruit valve and seed coat [92]. PYK10-binding protein 1 is expressed during the seedling development stage [93] and zein-binding domain-containing protein is similar to the *Zea mays* protein FLOURY 1 that is involved in protein body development and binds to zein proteins, which are seed storage proteins [94]. On the other hand, we identified two proteins related to mRNA processing: U2 small nuclear ribonucleoprotein A involved in intron recognition during pre-mRNA splicing in plants [95] and RNA-binding protein 1-like isoform X2, an alternative splicing (AS) regulator that binds to specific mRNAs and modulates auxin’s effects on the transcriptome [96], and a cyclin D5-3-like protein, that in maize was accumulated only during the early seed germination [97,98]. Surprisingly, cyclin showed the highest accumulation in mistletoe seeds. In orthodox seeds, the embryo is metabolically quiescent, this is why the resumption of essential processes, including transcription, translation, and DNA repair, is the first sign of germination [99,100]. There are diverse changes in protein accumulation between TI and T2 stages, the seed and start of germination, respectively. Cyclin-D5-3, RNA-binding protein, and zein-binding protein vanished in stage T2. However, stage T3 began the accumulation of a leucine-rich repeat-containing protein, that is critical to maintaining chromosome integrity [101]. Protein N-lysine methyltransferase can monomethylate Lys-20 of histone H4 to effect transcriptional repression of some genes [102] and 3-oxoacyl-acyl-carrier synthase involved in the fatty acid biosynthesis whose loss of function affects normal seedling development [103,104,105], which are accumulated proteins related to metabolism, transcription, and development. T4 and T5 stages contain similar proteins involved in phytohormone signaling such as tryptophan aminotransferase-related proteins involved in root development, and similar tissue such as the haustorium. In summary, the data suggest that *P. calyculatus* seeds like other recalcitrant seeds are metabolically active, and express hundreds or thousands of genes involved in seedling and tissue development [106].

### 3.3. Spatial Secretion of Cell Wall-Degrading Enzymes

Phytopathogenic microorganisms such as bacteria and fungi produce an arsenal of different enzymatic activities which, when combined, can degrade all lignocellulosic and protein components of a plant cell wall in order to invade and infect the host plant [107,108,109,110,111,112]. In that aspect, parasitic plants, during, host invasion, are capable of degrading lignocellulosic components and protein through the synthesis of cellulases, xylanases, glucanases, pectinesterases, and peptidases [1,113,114,115]. However, in *P. calyculatus*, that infects mesquite trees, there are no reports indicating that it secretes lignocellulolytic and peptidase enzymes. In that sense, we detected a group of proteins that were accumulated in the main infective stages such as pectinesterase 15 and DeSI-like (peptidase-like) protein that are likely cell wall-degrading and peptidase enzymes, respectively [52,53,54,80,81,116]. A pectin methylesterase enzyme was secreted from calli and seedlings of *Orobranche aegyptiaca* and *O. cumana* and it was speculated that the enzyme may have a function in host infection [51,117,118]. Additionally, a papain-like cysteine peptidase (cuscutain) has been identified in *Cuscuta reflexa*, involved in haustorium development and penetration [119,120]. The proteome revealed a metabolic enzyme related to glycoside hydrolases (pectinesterase) involved in complex polysaccharide degradation, and a DeSI-like protein similar to a peptidase, for which we performed measurements to obtain the different spatial activities of cellulases, xylanases, endo-1,4-β-glucanases, β-1,4-glucosidases, and peptidase-like proteins secreted by *P. calyculatus* during invasive stages on mesquite trees. Cellulase and β-1,4-glucosidase activities peaked in stage T3, which are active in haustorium development, whereas endo-1,4-β-glucanase, xylanase, and protease activities increased in stage T4, and remained constant in stage T5, and would have been active in the haustorium penetration/connection with the xylem of the host. According to our morphological observations and enzymatic activities, with which we associated the cell wall-degrading enzymes during germination, cellulases and β-1,4-glucosidases are the more active enzymes during haustorium penetration, and they probably degrade the host’s cortex. Endo-1,4-β-glucanase and xylanase are mainly involved in the degradation of internal cell tissues due to haustorium growth after having reached the xylem and cambium of the host branch. Depending on the host type, the enzymes can be activated asynchronously, for example, in *Cuscuta reflexa,* cellulases and xylanases are more active than glucosidase in the haustorial region [121]. During the invasive growth of *Cuscuta reflexa* in the host *Pelargonium zonale*, a high level of xyloglucan endotransglucosylase/hydrolase (XTH) activity was detected during host penetration [31]. The above explains how some enzymes are strongly activated in the penetration process of the host, such as in *P. calyculatus* infection.

A picture of the invasion mechanism of the haustorium penetration into the host could involve the activity of the first group of cellulase enzymes, which would degrade the cellulose surface layers of the cortex, and allow for secondary glucosidase activity. After these cortex-degrading activities, it is possible that the tip of the young haustorium can physically anchor on the pit cortex and begin to penetrate via pressure and cracking the bark cortex. However, those enzymes are subsequently used to continue penetration. A second set of enzymes such as glucanase and xylanase act in later stages such as stage T4 when the haustorium penetration into the host occurs and stage T5 when the haustorium completely connects with the xylem and the intruder tissue invades the host branch. Lastly, peptidases produced by the haustorium may be responsible for the breakdown-associated membrane proteins and lipoproteins from the host cell walls separating the host tissue to permit the penetration into the host, thereby liberating nutrients which are then available for growth and development of the haustorium tissue [122,123]. For example, cellulases in *Orobranche aegyptiaca* appear as the primary enzymes involved in establishing haustorial connection with the host root and, additionally, xylanases and proteases have a secondary function and might be involved in degrading xylose, proteins, and lipoproteins of the cell walls and membranes of the host tissues [35].

Particularly, cellulases are multi-enzyme complexes of different enzymes such as exoglucanase, endoglucanase, and beta-glucosidase that in combination with xylanases act synergistically for complete hydrolysis of cellulose and hemicellulose [124,125]. Cellulose fibers are firstly cleaved by endoglucanase, releasing small cellulose fragments with free reducing and non-reducing ends which are attacked by exoglucanase to release oligosaccharides and cellobiose that are hydrolyzed into glucose monomers by beta-glucosidase. Beta-glucosidase completes the final step of hydrolysis by converting the cellobiose (an intermediate product of cellulose hydrolysis) to glucose [125]. Xylanases are hydrolyases capable of breaking down hemicellulose into a mixture of xylooligosaccharides of different sizes [126]. Thus, it is probable that *P. calyculatus* haustorium cell wall-degrading enzyme secretion must be release in a specific order to degrade the three fundamental plant polymers. However, for this, a more in-depth study is needed to indicate which type of enzyme acts first. Taken together, our results demonstrate that those proteins could be a core of host invasion and have an important role in *P. calyculatus* seedlings, haustorium development, and/or host tree infection.

### 3.4. Metabolic Aspects during Germination

Mistletoes partially assimilate their own carbohydrates but also liberate carbohydrate and nitrogen compounds retrieved from host xylem [127]. In addition, during the seed-to-seedling transition of any seed, metabolism is maintained by protein catabolism [85], and photosynthetic transcripts are down-regulated in both parasitic and other plants [128]. Chlorophyll and nitrogen contents from seedling establishment to adult life stages have been quantified in other *Psittacanthus* and mistletoe species [127,129,130,131,132]. Protease activity is directly related to protein catabolism, which in consequence impacts in the nitrogen and chlorophyll content in seeds, germination, and seedling development in *P. calyculatus*, which are development programs previous to seedling establishment [82]. The protease activity increased from the T1 to T5 invasive stages. There was activity in stage T1 as the mistletoe seeds are recalcitrant and it is to be expected that metabolism was kept active. Then, with the mistletoe seed germination, protease activity increased to mobilize protein storage, chloroplast proteins were degraded to promote the radicle growth, and the chlorophyll concentration decreased dramatically in stage T5. Finally, there was another increase in T3–T5 stages due the penetration into host tissues to extract nutrients. Those data added to the changes in accumulation of proteins are related tocatabolism, highlighting the proteins’ role during early stages of infection of *P. calyculatus* not only as regulators but also as metabolic molecules.

### 3.5. Phytohormone Regulation of Germination and Seedling Establishment

Phytohormones such as auxin and cytokinin regulate a variety of physiological processes, including apical dominance, tropic responses, lateral root formation, and vascular differentiation in plant cells [133]. Parasitic plants modify the morphology and physiology of the host, and the synthesis of phytohormones [134], stimulating the production of phytohormones that regulate their growth as well as that of the host [41]. The haustorium is not an exception, there are reports that mention that phytohormones regulate the haustorium development [32,34,43,135].

We identified four auxins and six cytokinins in *P. calyculatus* infective stages. The auxin IAA-L-Glu peaked with a high concentration in T1, T2, and T3 stages, and its function is related to detoxification [136,137]. In contrast, IAA-L-Asp was quantified in high amounts in T4 and T5 stages and mistletoes leaves, which have a similar function for auxin deactivation [138]. Delavault [113] reported that the auxin regulation pathway is required in the early formation of the *Phtheirospermum japonicum* haustorium, as well as for the transport and utilization of nutrients, which suggests that in all mistletoe infective stages (T1–T5) there is an important regulation of IAA-L-Asp related to the formation of the haustorium and a correct use of nutrients, similar to that reported by Tomilov [139], who reported that in *Triphysaria versicolor* there is an accumulation of auxins in early stages of haustorium development and causes the parasite to generate specific mechanisms related to infection. Similarly, Zhang [37] found that in *Thesium chinense* there are gene overexpressions related to auxins in the haustorium zone, indicating the importance of those phytohormones in the haustorium development. This suggests that auxin conjugates such as IAA-L-Glu and IAA-L-Asp are present as a way to regulate IAA and so that mechanisms related to infection such as haustorium formation, transport, and utilization of host nutrients can develop correctly. In the *P. calyculatus* haustorium development of this study, IAA was detected in lower proportions in all infective stages, suggesting a basal expression of different genes related to IAA. This active form of auxin is responsible for the development of vascular tissue, initiation of the radicle, control of cell elongation and division, and transmission of environmental signals [140], that occur in parasites such as *Cuscuta reflexa*, and a relationship was found in the formation of an adhesive epithelium in tomato [141], as well as in the germination of *Phelipanche ramosa* and *Orobanche cumana* seeds, when they are in contact with exudates from tobacco and sunflower roots [38]. IAA-L-Ala was not detected in any of the mistletoe infective stages; this auxin is involved in cell expansion, suggesting that this phytohormone, involved in pathway regulation, is probably not required in *P. calyculatus* haustorium development or seedlings. Treatments with haustorium inductors up-regulated auxin biosynthesis genes, *Phelipanche ramose* initiated haustorium formation in the presence of cZ/tZ cytokinin species, and the inhibition of perception of cytokinin signaling blocks haustorium initiation [55,139,142,143]. Those effects may be related to the known induction of cell expansion and cell divisions by those two phytohormones [144]. Additionally, IAA is a natural auxin, which in conjugation with amino acids produces IAA-L-Asp, IAA-L-Glu, and IAA-L-Ala, allowing its storage [145]. In that sense, the auxin signaling is more active in seeds in comparison with other mistletoe invasive stages and leaves. The auxin storage (IAA-L-Asp and IAA-L-Glu) forms diminished in the seed, probably due to a high demand to coordinate development programs during germination and seedling establishment. The increase in IAA-L-Asp could be due to extraction from host tissues.

Cytokinins such as DZ were determined as having the highest proportion in infective stages T1, T2, and T3. This phytohormone is responsible for leaf growth and is highly present in dormant seeds, and is also related to the processes of endosperm expansion, germination, and nutrient allocation for growth and development of true leaves. The phytohormone cZR was present in a high concentration in T2 and T5 stages and in leaves. It is important to note that this phytohormone induces and promotes cell division, chlorophyll synthesis, and senescence delay [146]. *Phtheirospermum japonicum*, a parasitic plant that infects Arabidopsis roots, showed different levels of bioactive cytokinins that increased with a mobilization of this phytohormone towards the host, causing changes in cell responses, cell division, and differentiation, modifying root morphology, and negatively impacting the stability of the host [38]. The amount of active cytokinins such as cis-zeatins may also be related to the mistletoe infection process. Phytohormones such as kinetin and BA were not detected in any of the mistletoe infective stages. They are important for stimulating cell division and elongation and delaying senescence, respectively. tZ was detected in stage T2, this is the most important cytokinin given its high activity, it is responsible for vascular development and secondary xylem development, has a role in root/aerial part signaling, and may be related to the initial settlement of mistletoe and the subsequent penetration that was generated in stage T3. iP was not detected in T4–T5 stages and in leaves, this is a cytokinin related to the efficient use of water and is accumulated in the phloem, which correlates with the establishment of the mistletoe once the connection with the vascular system is made.

## 4. Conclusions

In our experiments, we analyzed the infection phenology of *P. calyculatus* mistletoe on mesquite (*P. laevigata*) host trees. Enzymatic analysis revealed the participation of enzymes that can penetrate the host tissue and are involved in establishment and uptake of nutrients. In that aspect, cellulase and β-1,4-glucosidase are active in haustorium development, whereas endo-1,4-β-glucanase, xylanase, and protease are active in haustorium penetration and connection with host xylem. All proteomics changes indicate a complex induction to achieve the successful growth of the parasite. Auxins and cytokinins are indicative of the importance of phytohormones in development and with special emphasis in parasite establishment. This study lays the bases for understanding the role of different biomolecules and their involvement in the infection process of mistletoe in mesquite, which in the future will allow the development of a control method.

## 5. Materials and Methods

### 5.1. Plant Material and Study Site

The collection of plant material was performed in a population of *P. laevigata* (mesquite) highly infected with *P. calyculatus* in a suburban area of Irapuato in the state of Guanajuato in Central Mexico (20°43′ N; 101°19′ O at 1730 m a.s.l). Weather of the area is mainly dry (43% of the state), especially at the northern region, whilst the remaining region is mild and humid. The annual average temperature is 18 °C, the highest is 30 °C, and the lowest is 5.2 °C in January, and a rainy season occurs in the summer.

### 5.2. Initial Infection Stage Collection

Different development stages were identified based to the observations of [26] to identify the life cycle scheme in *P. calyculatus*, but they were not collected due to their high amount of lignocellulose. For the identification of germination and infection stages, mature fruits were collected, and their exocarp was manually removed and seeds were placed on the host branches. Seeds contain a green viscin that works like a glue and adheres the seed tightly on the branch. The first five stages were collected, mature and sticky seed inoculated on the bark (T1), seed establishment that presented a polycotyledonous embryo (T2), cotyledon aperture (like a star) (T3), haustorium formation and penetration (T4), haustorium connection and vegetative growth (T5). Each infective stage was examined by passing a sterile scalpel between the tissue and the bark of the host branch. Except in T4–T5 stages, the tip of the scalpel was used as a shovel to scarify and remove the haustorium tissue found within the host branch. Those samples included green vegetative material of polycotyledonous embryo and the first true leaves (T4–T5). For comparative samples between infective stages, young and healthy leaves without visible infection by pathogens, after discarding the petiole, were used as controls. All tissues were flash frozen with dry ice and stored at −80 °C in an ultra-low freezer (Thermo Scientific, San Fernando, CA. USA) until experimental use.

### 5.3. Total Protein Extraction of P. calyculatus Stages and Leaves

Protein extraction was based on the protocol of [83] with some modifications. Tissues were ground in liquid nitrogen, and 0.2 g of sample was placed in 1 mL 10% TCA/acetone and centrifuged at 12,000 rpm for 3 min at 4 °C. The samples were washed with 80% methanol/0.1 M of ammonium acetate and centrifuged and washed in 80% acetone, centrifuged as before, and resuspended in a mixture (1:1) of 0.4 mL of phenol (Tris-buffered, pH 8.0; Sigma St. Louis, MO, USA) and 0.4 mL of dense SDS buffer (30% sucrose, 2% SDS, 0.1 M Tris-HCl, pH 8.0, and 5% β-mercaptoethanol). The mixture was vortexed for 5 min and then centrifuged at 12 000 rpm at room temperature for 5 min. The phenol phase was recovered and 0.4 mL of fresh SDS buffer was added twice and processed as before. Tubes that contained the phenol phase were filled with 80% methanol/0.1 M of ammonium acetate, stored at –20 °C for 30 min, and centrifuged for 5 min. Pellets produced by precipitation were washed twice with 80% ethanol. The protein content was determined using the Bradford kit (Bio-Rad, Hercules CA) with bovine serum albumin (BSA) as standard [147].

### 5.4. SDS-PAGE

The electrophoretic separation of proteins was performed by discontinuous 12% SDS–polyacrylamide gel electrophoresis (PAGE) [148]. Extracts of 25 µg of protein per sample were loaded into the wells and were separated on a vertical dual mini gel electrophoresis device (Mini-PROTEAN^®^ Tetra Vertical Electrophoresis Cell, Bio-Rad, Hercules, CA) at 160 V and 60 mA. Gels were stained with Coomassie colloidal blue overnight and destained in 40%/100% methanol:acetic acid solution for 1 h. Bands were identified on a white background using an imaging system (Kodak Gel Logic Imaging System GL 112).

### 5.5. Two-Dimensional Electrophoresis (2-DE)

In order to analyze the proteome of different stages of *P. calyculatus*, proteins were subjected to 2-DE. Protein samples (60 μg) were re-suspended in 125 μL of rehydrating solution (7 M urea, 2 M thiourea, 2% CHAPS, 18 mM of dithiothreitol (DTT), 2% IPG buffer (pH 3–10 and 4–7)) and then each individual sample was loaded onto 7 cm immobilized pH gradient (IPG) dry strip gels (pH 4–7) and allowed to passively rehydrate for 16 h. Then, the isoelectrofocusing of IPG strips was carried out according to the following protocol at 50 μA max and 20 °C: step 1, step and hold at 500 V for 1 h; step 2, gradient 1500 V for 1 h; step 3, gradient 6000 V for 1.5 h; step 4, gradient 8000 V for 2.5 h. Prior to the two-dimensional electrophoresis, the strips were equilibrated two times each for 15 min under gentle shaking in 10 mL of SDS reduction buffer (50 mM Tris-HCl (pH 8.8), glycerol 30%, urea 6 M, SDS 3%, 1% DTT, and bromophenol blue 0.002%) and, subsequently, the strips were placed in SDS reduction buffer with iodoacetamide 4%/10 mL and incubated as in the previous step. The strips were fixed on a continuous 12% SDS-PAGE gel at 160 V and 60 mA. The gel was stained with Coomassie blue as previously described and subsequently developed. To compare the differences in protein accumulation of infective stages, images were first normalized by a weighted marker Euclidian vector approach, where the Euclidian distance was calculated for the vectors of the protein molecular weight markers of each gel. Afterwards, spot densitometry was carried out using the free processing program ImageJ, and differences in protein expression were calculated by dividing the densitometry units of each spot by the Euclidian norm calculated for its corresponding gel [149] and were represented in a heat map.

### 5.6. Protein in Gel Digestion, LC-MS/MS, and Data Analysis

Protein spots of interest were cut from the gel matrix and digested [150] to generate peptides, which were fractionated with a C18 Sep-Pak cartridge (Waters) before injection into a mass spectrometer. The cartridge were conditioned with 2 mL of acetonitrile (ACN), equilibrated with 3 mL of 10 mM of ABC. Next, peptides resuspended in 500 µL of 20 mM ABC were applied to the cartridge and washed with 3 mL of 10 mM of ABC. The peptides were eluted with 8%, 14%, 20%, 26%, 32%, and 50% ACN in 10 mM ABC. The six fractions were vacuum dried. All peptide fractions were resuspended in 0.1% formic acid in water and analyzed in an Easy Nano-Liquid Chromatography (LC)-1000 system coupled with a LTQ Orbitrap Elite mass spectrometer (Thermo Fisher Scientific). Peptides were loaded in the trap column (C18 PepMap100, 300 µm × 1 mm, 5 µm, 100 Å; Dionex) and separated in the column (Acclaim PepMap C18, 15 cm × 75 µm × 3 µm, 100 Å; Dionex) with a binary solvent system of 0.1% formic acid (A) and 0.1% formic acid in CAN (B) for 120 min. The linear gradient ranged from 5 to 50% B. The mass spectrometer was operated in positive ionization mode and in the top-15 data-dependent mode. Full MS scans were acquired using the Orbitrap analyzer with a mass range from 300 to 1800 m/z at a resolution of 60,000, using an automatic gain control (AGC) of 400,000 ions. Peptides were fragmented with a normalized collision energy of 30. MS/MS spectra were acquired with the ion trap using a width of 0.7 *m*/*z*, and a target value of 10,000 ions. Dynamic exclusion was set to 40 s and ions with a charge of +1 or greater than +7 were excluded from fragmentation. The data were processed utilizing Proteome Discoverer 2.2 (Thermo Fisher Scientific) and MaxQuant Mascot (Matrix Science) with the MS/MS search engine database of NCBI (http://w.w.w.ncbi.nlm.nih.gov/, updated on June 2022) employing the MS BLAST program [151]. Given that *P. calyculatus* is placed among the core eudicots in the Santalales order (Angiosperm Phylogeny Group) [152] we decided to generate a custom database containing protein sequences from Beta vulgaris, Arabidopsis thaliana, and *Solanum lycopersicum*.

### 5.7. Cellulase Activity

The enzymatic extraction was performed according to [35], with modifications. Briefly, samples of 50 mg of fresh weight were mixed with 500 μL of extraction buffer (sodium acetate 0.1 M pH 5.2, NaCl 10%), vortexed for 1 min, and centrifuged at 12,000 rpm for 30 min at 4 °C. The pellet was discarded, and the supernatant was stored at 4 °C until used. The activity of the cellulase enzyme was measured in filter paper units (FPU). For that, a 5 mm strip of a Whatman #41 filter paper was added to 20 μL enzyme extract containing 40 μL of 5 mM sodium acetate buffer, pH 4.8. The samples were incubated for 1 h at 50 °C. To measure released reducing sugars, mixtures were determined after stopping the hydrolysis by addition of 120 μL DNS solution and finally incubated at 95 °C for 5 min. After cooling, 36 μL of each mixture was transferred to a 96-well plate and 160 μL deionized water was added and samples were read at 540 nm. A glucose standard curve was made in order to convert the absorbance values into reducing sugar concentration released by the enzymatic hydrolysis.

### 5.8. Endo-1,4-β-glucanase

Endo-1,4-β-glucanase-like activity was measured following the protocol by [153] with modifications. A volume of 30 μL of 2% (*w*/*v*) carboxymethylcellulose (CMC) as substrate (prepared in sodium acetate buffer, pH 4.8) was mixed with 30 μL of enzyme extract. The mixtures were incubated for 30 min at 50 °C, and 60 μL of DNS was added and incubated at 95 °C for 5 min. Then, samples of 100 μL were transferred to a 96-well plate, and the absorbance was measured at 540 nm. One enzyme unit was defined as the amount of enzyme capable of releasing 1 μmol of reducing sugar per minute.

### 5.9. β-glucosidase Activity

Enzymatic activity was measured in a 96-well plate with the β-Glucosidase Activity Assay Kit (Sigma-Aldrich, St. Louis Missouri, USA, MAK129) according to the manufacturer’s instructions. P-Nitrophenyl β-D-glucopyranoside (β-NPG) was used as a substrate (1 mM) and absorbances were measured at 405 nm at the beginning of the reaction, and after a 20 min incubation at 37 °C. The number of units per liter (U/L) was measured. One unit of β-1,4-glucosidase is the amount of enzyme that catalyzes the hydrolysis of 1.0 μmole substrate per minute at pH 7.0.

### 5.10. Xylanase Activity

Xylanase-like activity was measured following the protocol of [154] with some modifications. Thirty microliters of enzymatic extract was mixed with 30 μL of xylan (1.0%) from beechwood (prepared in sodium acetate buffer, pH 5.0) as substrate. The plate was incubated for 40 min at 50 °C. After this time, 120 μL of DNS was added to each tube and heated at 95 °C for 5 min. Thirty-six microliters of the solution from each tube was transferred to a 96-well plate, 160 μL of deionized water was added, and the absorbances were read at 540 nm. One unit is interpreted as 1 μmol of xylose equivalents released per minute under the conditions of reaction used.

### 5.11. Proteolytic Activity

Soluble proteins were extracted by grinding 100 mg of infective stage samples in 400 μL ice-cold 0.1 M Tris-HCl (pH 7.4) and incubated for 15 min at 4 °C in a mixer; after this time, samples were centrifuged (12,000 rpm) for 15 min at 4 °C and the resulting extracts were collected and stored at −70 °C until use [83]. Then, the protease activity was estimated using the protocol of [35]. In short, 20 µg of protein extract was combined with 100 µL buffer (0.1 M Tris-HCl, pH 7.4) and pre-incubated for 15 min at 37 °C. After this time, 1% BSA solution was added and incubated for 1 h at 37 °C. The reaction was stopped by adding 200 µL cold trichloroacetic acid (5.0%) and incubated for 30 min at 4 °C. Samples were centrifuged at 12,000 for 20 min, supernatants were diluted, and finally absorbance was read at 280 nm.

### 5.12. Chlorophyll Content

The chlorophyll content was measured in the different mistletoe infective stages based on the protocol of [155]. The pigment was extracted from 100 mg of tissue, which was homogenized in 500 µL of ethanol (96%) for 24 h in darkness at 4 °C. After this time, samples were centrifuged for 15 min at 12,000 rpm. Supernatant absorbance readings for the extracts were obtained at 666 nm (chl a) and 653 nm (chl b), and chlorophyll a + b contents and their ratios were calculated using (7.15 × A663.2) + (18.71 × A646.8)]/[1000 x (fresh weight of leaves), and reported as mg Chl. per g FW.

### 5.13. Phytohormone Determination

Samples of 200 mg of lyophilized tissue were resuspended in 1 mL of extraction buffer consisting of methanol:water:formic acid (15:4:1, v/v/v) supplemented with isotopically labeled 13C-[13C6]indole-3-acetic acid (13C-IAA) and [2H6]N6-isopentenyladenine (D-iP) as standards for auxin and cytokinin, respectively. After solids were removed by centrifugation at 15,000 rpm for 15 min at 4 °C, the supernatant was transferred to a fresh tube and the residue was re-extracted for 30 min in an additional 500 µL extraction mixture at −20 °C and centrifuged again. The supernatants were combined and evaporated in a SpeedVac at 10 mBar and 40 °C to ¼ of the initial volume. Plant extracts containing isotopically labeled standards were diluted in 1 mL of SPE load solvent (1 M formic acid in water) and passed through an SPE Oasis MCX cartridge (Waters Co, Mississauga, ON, Canada), which had been previously equilibrated with 1 mL of methanol, followed by 1 mL SPE load solvent. Auxins were eluted with 1.5 mL of SPE elute 1 solvent (100% methanol) after the column was washed with 1 mL of water. Subsequently, the cytokinins were eluted with 1.5 mL of SPE elute 2 solvent (0.35 M ammonium hydroxide in 70% methanol). SPE elutes were evaporated in a SpeedVac at 10 mBar and 40 °C to dryness. The fractions were stored at −20 °C until LC-MS analysis. LC-MS/MS analysis was performed using a Thermo LTQ Orbitrap, equipped with a heated electrospray ionization (HESI-II) source with sheath gas set to 60, auxiliary gas set to 20, source temperature set to 310 °C, and spray voltage 4 kV in a positive mode. Chemical fragmentation of auxins and cytokinin were as previously reported [156]. Chromatographic separations were performed using a 17-reverse phase ZORBAX Eclipse XDB C18 column (150 X 4.6 mm i.d., 5 µm particle size, 80 Å pore size, Waters Co, Mississauga, Ontario, Canada). A gradient of 0.1% formic acid in water (A) and 0.1% formic acid in acetonitrile (B) was used during LC separations. A flow rate of 0.3 mL min −1 was used, and 2 µL of the injection volume was used. Phytohormones relative quantitation was determined by the area under the curve using Qual Browser and Quan Browser implemented in Xcalibur version 4.1 (Thermo Fisher Scientific) generated by LC-MS/MS peaks [157].

### 5.14. Statistical Analysis

Enzymatic activities and phytohormone levels were expressed as means ± standard deviation and analyzed using global LSD (Least Significant Difference) post hoc tests after univariate analysis of variance (ANOVA) using Statistical Package for the Social Sciences 17.0 (SPSS Inc., Chicago, USA). 

## Figures and Tables

**Figure 1 plants-12-00464-f001:**
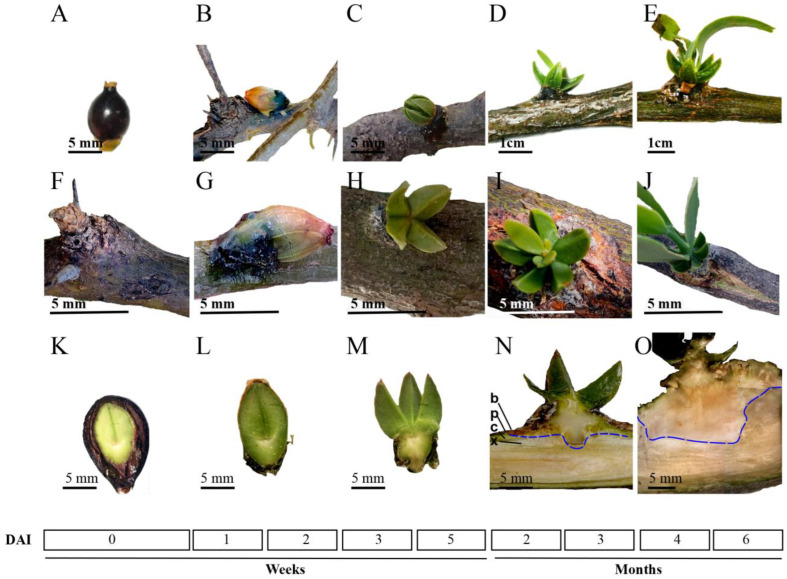
Infective stages of *P. calyculatus* on the mesquite host tree branch. Representative images were analyzed according to phenological shoot characteristics: (**A**) T1 stage corresponds to mature seed/fruit; (**B**) T2 stage, germination/holdfast formation; (**C**) T3 stage, seedling establishment/haustorium activation; (**D**) T4 stage, haustorium penetration; and (**E**) T5 stage, haustorium connection with the xylem. Bark of host without mistletoe seed (**F**); seed viscin on the bark (**G**); dry viscin fusion with bark (**H**); bark breakdown (**I**); and haustorium bulge formation (**J**). Transversal dissection of each invasive stage: T1 (**K**), T2 (**L**), T3 (**M**), T4 (**N**), and T5 (**O**). The scale bar in each picture is indicated. DAI: days after inoculation. b: bark, p: phloem, c: cambium, and x: xylem. Dashed blue line indicates the limit between haustorium and host tissues.

**Figure 2 plants-12-00464-f002:**
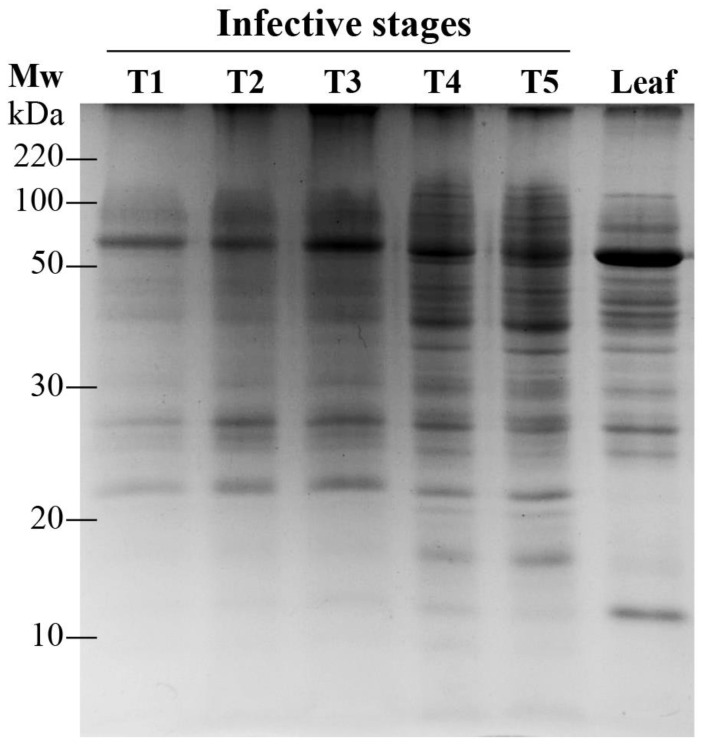
Proteomes obtained by SDS-PAGE for *P. calyculatus* infective stages.

**Figure 3 plants-12-00464-f003:**
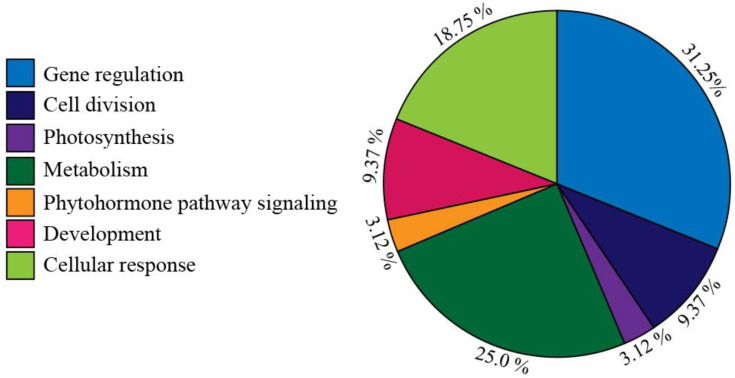
Functional protein categories of *P. calyculatus* infective stages. Pie chart illustrating the major functional proteins.

**Figure 4 plants-12-00464-f004:**
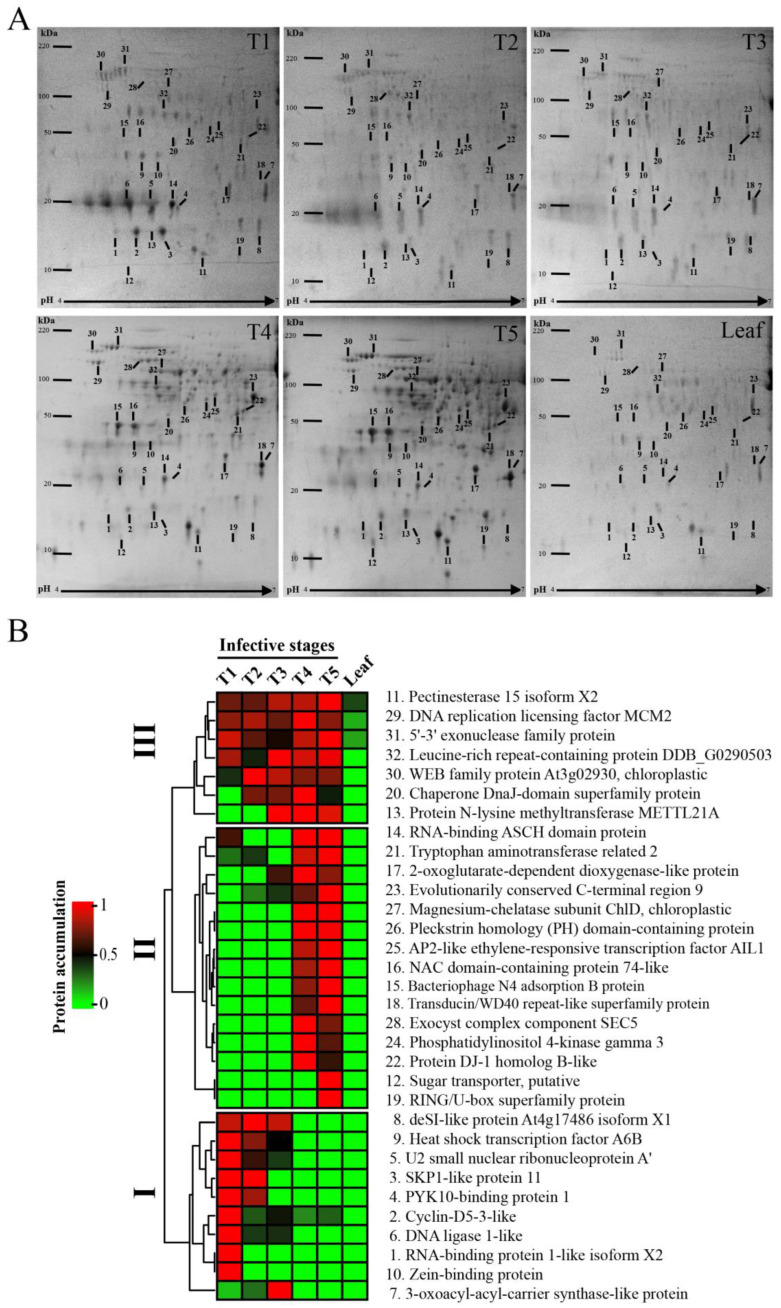
Proteome and heat map analysis of *P. calyculatus* infective stages. (**A**) Protein profiles and (**B**) heat map cluster analysis of the protein accumulations of the different infective stages (T1–T5) compared with mistletoe leaves.

**Figure 5 plants-12-00464-f005:**
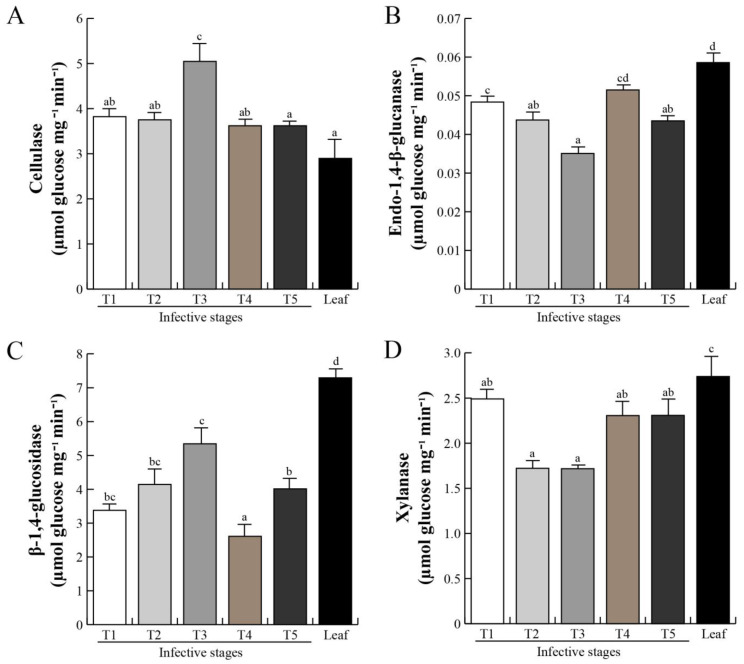
Cell wall-degrading enzyme activities from infective stages. The enzymatic activities were assayed with total protein extracts from the *P. calyculatus* infective stages and leaves. (**A**) Cellulase, (**B**) endo−1,4−β−glucanase, (**C**) β−1,4−glucosidase, and (**D**) xylanase. Bars represent mean ± SE (n = 10), different letters above the bars indicate significant differences among infective stages (*p* < 0.05 according to ANOVA and Tukey test).

**Figure 6 plants-12-00464-f006:**
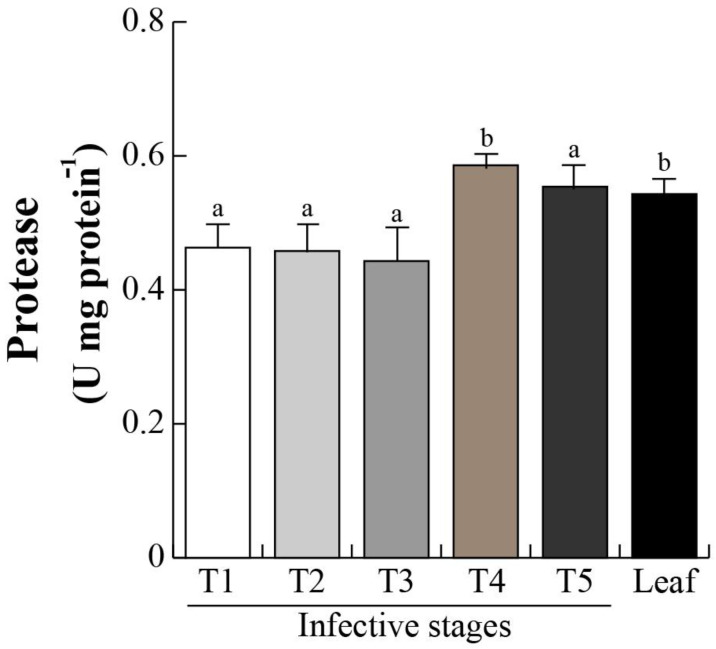
Protease activity of infective stages. Protease activity was measured from total seed protein extracts from *P. calyculatus* infective stages and leaves. Bars represent mean ± SE (n = 10), different letters above the bars indicate significant differences among infective stages (*p* < 0.05 according to ANOVA and Tukey test).

**Figure 7 plants-12-00464-f007:**
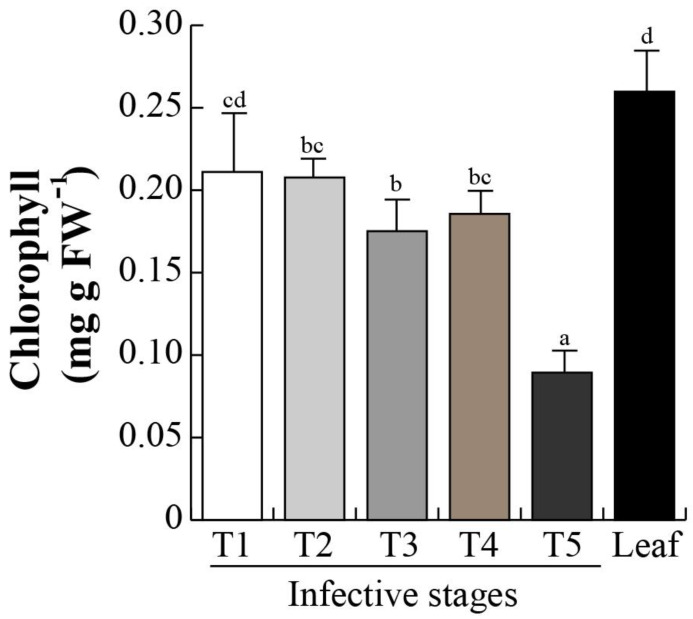
Chlorophyll content of infective stages. The pigment was extracted from each invasive stage. Bars represent mean ± SE (n = 10), different letters above the bars indicate significant differences among infective stages (*p* < 0.05 according to ANOVA and Tukey test).

**Table 1 plants-12-00464-t001:** Proteins detected in *Psittacanthus calyculatus* infective stages.

Spot	Description	Accesion	MS BLAST	Organism	Function	Mw	pI
			Score			kDa	
**Gene regulation**							
1	RNA-binding protein 1-like isoform X2	XP_010322224.1	412	* Solanum lycopersicum *	Translation, ribosomal structure and biogenesis	22.2	4.4
5	U2 small nuclear ribonucleoprotein A′	XP_004235867.1	491	* Solanum lycopersicum *	Nuclear mRNA splicing	32.3	5.5
6	DNA ligase 1-like	XP_009765758.1	297	*Nicotiana sylvestris*	DNA recombination and DNA repair	55.0	5.0
9	Heat shock transcription factor A6B	NP_188922.1	759	*Arabidopsis thaliana*	DNA binding	46.7	5.0
13	Protein N-lysine methyltransferase METTL21A	XP_004244751.1	549	*Solanum lypersicum*	Methyltransferase of heat shock protein 70 (HSP70) family members	31.7	4.9
14	RNA-binding ASCH domain protein	NP_001324874.1	449	*Arabidopsis thaliana*	RNA-binding domain during coactivation	27.4	5.2
23	Evolutionarily conserved C-terminal region 9	NP_174117.2	1077	*Arabidopsis thaliana*	RNA binding	61.2	6.3
25	AP2-like ethylene-responsive transcription factor AIL1	NM_001366759.1	1080	*Solanum lypersicum*	DNA binding	62.0	6.0
29	DNA replication licensing factor MCM2	XP_004250699.2	1854	* Solanum lycopersicum *	3′-5′ DNA helicase/THO complex	108.1	5.1
31	5′-3′ exonuclease family protein	NP_001325748.1	2779	*Arabidopsis thaliana*	Exonuclease that digests recessed strands of DNA duplexes in the 3′ to 5′ direction	168.8	4.9
**Cell division**							
2	Cyclin-D5-3-like	XP_019069003.1	316	* Solanum lycopersicum *	Cell-cycle	24.5	4.7
30	WEB family protein At3g02930, chloroplastic	XP_004235278.1	1399	*Solanum lycopersicum*	Microtubule motor activity	109.9	4.9
32	Leucine-rich repeat-containing protein DDB_G0290503	XP_010674981.1	1019	*Beta vulgaris subsp. vulgaris*	Probable component of the transverse filaments	71.6	5.4
**Photosynthesis**							
27	Magnesium-chelatase subunit ChlD, chloroplastic	XP_004236627.1	1327	*Solanum lypersicum*	Chlorophyll biosynthesis	83.9	5.4
**Metabolism**							
3	SKP1-like protein 11	XP_010321926.2	368	*Solanum lypersicum*	E3 ligase complex	21.0	5.0
7	3-oxoacyl-acyl-carrier synthase-like protein	NP_196072.1	148	*Arabidopsis thaliana*	Fatty acid synthesis	32.9	6.2
8	deSI-like protein At4g17486 isoform X1	XP_010325508.1	401	*Solanum lypersicum*	Thiol peptidase	24.9	6.2
10	Zein-binding protein	NP_200591.2	786	*Arabidopsis thaliana*	Zein-binding domain-containing protein	44.0	5.2
11	Pectinesterase 15 isoform X2	XP_048504660.1	101	*Beta vulgaris subsp. vulgaris*	Acts in the modification of cell walls via demethylesterification of cell wall pectin	15.6	5.9
12	Sugar transporter, putative	NP_569015.1	261	*Arabidopsis thaliana*	Sugar transporter	18.2	5.2
19	RING/U-box superfamily protein	NP_196267.1	309	*Arabidopsis thaliana*	E3 ubiquitin ligase	22.0	6.4
20	Chaperone DnaJ-domain superfamily protein	NP_174319.1	650	*Arabidopsis thaliana*	Folding and degradation of proteins	50.7	5.3
**Phytohotmone pathway signaling**							
21	Tryptophan aminotransferase related 2	NP_567706.1	887	*Arabidopsis thaliana*	Trytophan aminotransferase/C-S-lyase	50.0	6.1
**Development**							
18	Transducin/WD40 repeat-like superfamily protein	NP_851281.1	675	*Arabidopsis thaliana*	Regulation of dynamic multi-subunit complexes	36.1	6.2
24	Phosphatidylinositol 4-kinase gamma 3	XP_010671446.1	1150	*Beta vulgaris subsp. vulgaris*	Phosphatidylinositol 3-and 4-kinase	63.8	5.7
28	Exocyst complex component SEC5	XP_010680634.1	2079	*Beta vulgaris subsp. vulgaris*	Involved in polarized cell growth	122.8	5.6
**Cellular response**							
4	PYK10-binding protein 1	NP_001030710.1	530	*Arabidopsis thaliana*	Inhibitor-type lectin that may regulate the correct polymerization of BGLU23/PYK10 upon tissue damage	32.1	5.8
15	Bacteriophage N4 adsorption B protein	NP_569035.1	104	*Arabidopsis thaliana*	Immune response	72.0	5.0
16	NAC domain-containing protein 74-like	XP_016449206.1	296	*Nicotiana tabacum*	Transcription activator involved in heat and endoplasmic reticulum (ER) stress responses	44.4	5.3
17	2-oxoglutarate-dependent dioxygenase-like protein	NP_001322930.1	653	*Arabidopsis thaliana*	Glucosinolates biosynthesis	37.7	6.3
22	Protein DJ-1 homolog B-like	XP_010675408.1	897	*Beta vulgaris subsp. vulgaris*	Glyoxalase I activity	47.1	6.3
26	Pleckstrin homology (PH) domain-containing protein	NP_850155.1	831	*Arabidopsis thaliana*	Binds specifically to phosphatidylinositol 3-phosphate (PtdIns3P)	56.0	5.9

**Table 2 plants-12-00464-t002:** Phytohormone level of *Psittacanthus calyculatus* during infective stages.

			Infective Stages				
Phytohormone	Abbreviation	T1	T2	T3	T4	T5	Leaf
**Auxins**							
Indole-3-Acetic Acid	IAA	1.3 ± 0.2 c	0.2 ± 0.01 a	ND	0.6 ± 0.1 b	0.3 ± 0.02 a	ND
Indole-3-Acetyl-L-Aspartic Acid	IAA-L-Asp	71.8 ± 10.3 c	8.0 ± 3.4 ab	7.0 ± 0.4 b	3.9 ± 0.1 a	6.1 ± 0.79 b	2.9 ± 0.3 a
Indole-3-Acetyl-L-Glutamic Acid	IAA-L-Glu	913.6 ± 170.9 e	121.9 ± 26.8 d	90.1 ±5.2 c	2.1 ± 0.1 b	2.7 ± 0.3 b	0.5 ± 0.05 a
Indole-3-Acetyl-Alanine	IAA-L-Ala	ND	ND	ND	ND	ND	ND
**Cytokinins**							
Dehydrozeatin	DZ	1.3 ± 0.2 b	3.2 ± 0.6 c	0.4 ± 0.03 a	ND	ND	ND
N6-furfuryladenine	Kinetin	ND	ND	ND	ND	ND	ND
N6-Isopentenyladenine	iP	0.1 ± 0.01 a	2.6 ± 0.34 b	0.2 ± 0.01 a	ND	ND	ND
N6-benzyladenine	BA	ND	ND	ND	ND	ND	ND
cis-Zeatin-Riboside	cZR	0.2 ± 0.03 b	0.9± 0.2 c	0.2 ± 0.02 b	ND	0.9 ± 0.15 c	0.08 ± 0.01 a
trans-Zeatin	tZ	ND	0.5 ± 0.1 a	ND	ND	ND	ND

Letters represent statistically significant differences for each phytohormone per infective stage (*p* < 0.05 according post hoc Tukey’s HSD after ANOVA). ND, not detected.

## Data Availability

Not applicable.

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
