# Peer review of "Protein Profiling of Psittacanthus calyculatus during Mesquite Infection"

_plants, 2023, doi:10.3390/plants12030464_

Round 1
Reviewer 1 Report
Line 53: delete "them"
Line 95: missing "it' after "Therefore,"
Line 113: write the name of the author before "[35]"
Line 114: write "establishment" instead of "established"
Line 116: write "monitor" instead of "monitoring"
Line 154: Fig. 3 should be Fig. 4
Line 164-171: The percentage is different from Fig. 3
Line 177: peptidase is different from pectinesterase as shown in Fig 4B
Line 179: "them" replace with "are"
Line 185: Fig. 5B should be Fig. 4B
Line 186: zein-binding protein is not No. 13
Line 190: peptidase is different from pectinesterase as shown in Fig 4B
Line 218: pectinesterase is not a proteolytic enzyme. The proteolytic enzyme is for protein while pectin is a carbohydrate
Line 676: peptidase doesn't belong to glycoside hydrolase
Author Response
Response: Thanks for analyzed our research.
Line 53: delete "them"
Response: The word was deleted, and you can check in L55, P2.
Line 95: missing "it' after "Therefore,"
Response: The word was added you can check in L97, P2.
Line 113: write the name of the author before "[35]"
Response: The author was added, and you can check in L115, P3.
Line 114: write "establishment" instead of "established"
Response:The word was added you can check in L116, P3.
Line 116: write "monitor" instead of "monitoring"
Response:The word was modified you can check in L118, P3.
Line 154: Fig. 3 should be Fig. 4
Response:The number was corrected, you can check in L207, P5.
Line 164-171: The percentage is different from Fig. 3
Response: The percentage were corrected based on the correct figure, you can check in L238-246, P5.
Line 177: peptidase is different from pectinesterase as shown in Fig 4B
Response: The suggestion was corrected, you can check in L311, P7.
Line 179: "them" replace with "are"
Response:The word was added you can check in L313, P7.
Line 185: Fig. 5B should be Fig. 4B
Response:The number was corrected, you can check in L319, P7.
Line 186: zein-binding protein is not No. 13
Response:The number was corrected, you can check in L320, P7
Line 190: peptidase is different from pectinesterase as shown in Fig 4B
Response: sentence was corrected, you can check in L324, P7.
Line 218: pectinesterase is not a proteolytic enzyme. The proteolytic enzyme is for protein while pectin is a carbohydrate
Response: Thanks for analyzed our mistake, more detail were added, and the sentences were corrected. Check L510-511, P10.
Line 676: peptidase doesn't belong to glycoside hydrolase
Response: Thanks for analyzed our mistake, more detail were added, and the sentences were corrected. Check L758-775, P15-16.

Reviewer 2 Report
See attached

Author Response
The objective of the work is to characterize the different stages of infection of Psittacanthus calyculatus a mistletoe with its host Mesquite (Prosopis laevigata) and to quantify the metabolites (auxins, cytokinin, chlorophyll) during the infection as well as the enzymes linked to the degradation of the cell wall (cellulase, xylanases, etc.). The manuscript is clear and the results are relevant.
Response: Thanks for analyzed our research.
Many references cited are not from the last 5 years. For example in the introduction we have 7 references that are more than 20 years old. In the discussion there are 8 references that are over 5 years old, 14 are over 10 years old and 12 are over 20 years old.
Response: Thank you for highlighting this important fact. The citations that we use are based on the direct impact they have on this research, the studies are few, but they are related in certain aspects (few enzymes, proteomes, only in parasitic plants), however, a few of those studies are classic reports of more than 10, 15, or more years, which we have to cite and are classic in this area. However, in the introduction we add 15 citations and in the discussion section 14 citations from years between 2020-2023.
Not all references are current.
Response: Thank you for highlighting this important fact. The citations that we use are based on the direct impact they have on this research, the studies are few, but they are related in certain aspects (few enzymes, proteomes, only in parasitic plants), however, a few of those studies are classic reports of more than 10, 15, or more years, which we have to cite and are classic in this area. However, in the introduction we add 15 citations and in the discussion section 14 citations from years between 2020-2023.
All figures, tables are appropriate with data that is well presented and easy to understand. But in figure 4 the title is not readable.
Response: Thanks for analyzed this part, for more readable, the figure title and presentation was modified, you can check in P8 and Lines:416-418.
Some references cited in the text do not appear as bibliographic references. For example: Lehmann et al 1995, Pio et al 2020, Wu and Xue 2010 are not in the references.
Response: The cites of those authors and missing cites were added and modified in text and literature section. Please, check L:746-753, P:15
The conclusion is not clearly reflected in the document.
Response: Thanks for check this part. The missing conclusion you can check in L919-932 in P :18-19.
Comments: Clear title
Response: Thanks for analyzed our research.
Abstract: Clear context and problematic as well as the results, but the methodology is missing in the abstract. You said you described 5 phenological stages of P. calyculatus but you did not say how you described them.
Response: Thanks for analyzed this part; we added some lines in this part. Please, check L21-22, P1.
Line 29-30: and we suggest that is a general infection mechanism for another mistletoe species. Replace the suggested word in the phrase these are the results.
Response: The suggestion was added. Please check L:29-31, P:1.
Introduction: 24 references, 7 of which are over 20 years old
Response: Suggestions were modified, corrected and added more updated citations
Results: The first paragraph looks like an introduction. Instead presenting of presenting the results obtained, Authors focused on describing P. calyculathus. In the second paragraph of this document, you are still describing the different stages of infection instead presenting the result that you obtained.
Response: Thanks for check this part. In that paragraph we are describing the part that we saw that nobody has done and where we took the tissues that were analyzed, we think it is important to briefly describe where we started from : (Geil et al. 2002), since it was the basis. Additionally, as we know, this manuscript is long in description and discussion, and we believe that it would help the reader in the context from which it started.
Discussion: Line 605, 651, 653: you should not write the names of the authors. Comply with guidelines
Response: The cites of those authors and missing cites were added and modified in text and literature section. Please, check L:746-753, P:15
Conclusion: the conclusion is missing
Response: Thanks for check this part. The missing conclusion you can check in L919-932 in P :18-19.
